Japanese knotweed (Fallopia japonica): an analysis of capacity to cause structural damage (compared to other plants) and typical rhizome extension

Fennell Mark 1 mark.fennell@aecom.com
Wade Max 1
Bacon Karen L. 2
1 AECOM , Cambridge , UK
2 School of Geography, University of Leeds , Leeds , UK
Martinelli Luiz
Electronic publication date: 2018 Jul 25
Publication date: 2018
Volume: 6
Electronic Location ID: e5246
Received 2018 Mar 26; Accepted 2018 Jun 26
Copyright: © 2018 Fennell et al.
Copyright year: 2018
Copyright holder: Fennell et al.
License: This is an open access article distributed under the terms of the Creative Commons Attribution License, which permits unrestricted use, distribution, reproduction and adaptation in any medium and for any purpose provided that it is properly attributed. For attribution, the original author(s), title, publication source (PeerJ) and either DOI or URL of the article must be cited.
License URL: https://creativecommons.org/licenses/by/4.0/

Keywords: Japanese knotweed, Fallopia japonica, Invasive species, Impacts, Structural damage, Rhizome

Funding: The authors received no funding for this work.

==============================
Fallopia japonica (Japanese knotweed) is a well-known invasive alien species in the UK and elsewhere in Europe and North America. The plant is known to have a negative impact on local biodiversity, flood risk and ecosystem services; but in the UK it is also considered to pose a significant risk to the structural integrity of buildings that are within seven m of the above ground portions of the plant. This has led to the presence of the plant on residential properties regularly being used to refuse mortgage applications. Despite the significant socioeconomic impacts of such automatic mortgage option restriction, little research has been conducted to investigate this issue. The ‘seven-m rule’ is derived from widely adopted government guidance in the UK. This study considered if there is evidence to support this phenomenon in the literature, reports the findings of a survey of invasive species control contractors and property surveyors to determine if field observations support these assertions, and reports a case study of 68 properties, located on three streets in northern England where F. japonica was recorded. Additionally, given the importance of proximity, the seven-m rule is also tested based on data collected during the excavation based removal of F. japonica from 81 sites. No support was found to suggest that F. japonica causes significant damage to built structures, even when it is growing in close proximity to them and certainly no more damage than other plant species that are not subject to such stringent lending policies. It was found that the seven-m rule is not a statistically robust tool for estimating likely rhizome extension. F. japonica rhizome rarely extends more than four m from above ground plants and is typically found within two m for small stands and 2.5 m for large stands. Based on these findings, the practice of automatically restricting mortgage options for home buyers when F. japonica is present, is not commensurate with the risk.

Introduction

Japanese knotweed (Fallopia japonica) is a tall, herbaceous, perennial plant with woody rhizomes when mature. F. japonica is now recognised as one of the most problematic weeds in the UK and Ireland (Environment Agency, 2013; Property Care Association (PCA), 2018). It is also recognised as one of the worst invasive alien species (IAS) at a European scale (Nentwig et al., 2017) and globally (Lowe et al., 2000), being particularly invasive in parts of North America, Europe, Australia and New Zealand (Centre for Agriculture and Bioscience International (CABI), 2018a). On a global scale its reputation as a problematic IAS primarily stems from its vigorous growth and impacts on riparian habitats (Child & Wade, 2000) coupled with difficulty of eradication (Bailey, 2013; Jones et al., 2018). Verified impacts include the creation of dense monodominant stands (Gillies, Clements & Grenz, 2016; Michigan Department of Natural Resources (MDNR), 2012); reductions in ecosystem services in riparian zones, for example by impeding access (Environment Agency, 2013; Gerber et al., 2008; Kidd, 2000; Urgenson, 2006); negative effects on native plant and invertebrate assemblages in riparian habitats (Gerber et al., 2008); reductions in species richness (Aguilera et al., 2010; Hejda, Pyšek & Jarošík, 2009; Urgenson, 2006) and abundance of native understory herbs, shrubs and juvenile trees in riparian woodlands (Urgenson, 2006); modifications to nutrient cycles (Urgenson, 2006); and impacts on flood defence through impeding water flow and facilitation of riverbank erosion (Booy, Wade & Roy, 2015; Environment Agency, 2013; Kidd, 2000).

The plant is associated with significant economic impacts in the UK, particularly in the development sector, due in large part to soil containing the species being classified as controlled waste, which can result in significant waste management costs (Williams et al., 2010; Pearce, 2015). Economic impacts have been estimated at £166,000,000 per year (Williams et al., 2010) in the UK; however, the validity of this, frequently misquoted, figure is strongly debated (Pearce, 2015).

Fallopia japonica was introduced to Europe from Japan in the mid-19th century by the Bavarian Phillip von Siebold, a renowned importer of exotic plants at this time (Bailey, 2013). In 1850, von Siebold sent a package to Kew Gardens in London, which included a female (male sterile) F. japonica plant (Bailey, 2013). Once established in Kew Gardens it was distributed throughout the UK, being planted in Victorian parks and gardens (Bailey, 2013). Despite rumblings from Victorian gardeners as far back as 1898, for example William Robinson (Bailey & Conolly, 2000), about the plant’s invasiveness, it was available for sale in UK nurseries up until at least 1990 (Philip, 1990). It was first recorded outside cultivation in South Wales in 1886 (Storrie, 1886) and is currently recorded in most hectads within the UK and Ireland (Botanical Society of Britain and Ireland (BSBI), 2018; Fig. 1A).

Figure 1 Distributions maps showing F. japonica records and soil shrink-swell potential.

(A) Records from the Botanical Society of Britain and Ireland live database based on presence/absence data in each hectad. Almost all hectads report fewer than 100 records. Map was produced using records collected mainly by members of the Botanical Society of Britain and Ireland (BSBI) (2018). (B) British Geological Society map showing areas at risk of shrink-swell action. Reproduced with the permission of the British Geological Survey ©UKRI. All rights reserved (British Geological Survey (BGS), 2018).

By the late 1970s, the invasive nature of F. japonica was becoming widely recognised (Bailey, 2013) in the UK (also see ‘Study species: Fallopia japonica’ below). Within the popular press and through various online sources, F. japonica is increasingly sensationalised and is credited on a regular basis with an ability to ‘grow through concrete’ and ‘destroy building foundations’ (Ellery, 2016; Sweeny, 2017; Willey, 2018). Accordingly, in the 21st century, property surveyors and lenders started taking an increasingly risk-averse approach to the species (Royal Institution of Chartered Surveyors (RICS), 2012). Ultimately, this has led to the presence of F. japonica on or near a residential property preventing its sale (Royal Institution of Chartered Surveyors (RICS), 2012; Pearce, 2015). Frequently, financial institutions will automatically restrict mortgage options where F. japonica is within the boundary of the property or within seven m of a habitable space, conservatory or garage. This ‘seven-m rule’ is derived from widely adopted government guidance, which states that F. japonica rhizome may extend seven m laterally from a parent plant (Environment Agency, 2013).

Where F. japonica is preventing a property sale, this issue can typically be resolved if evidence can be provided to a lender that an appropriate treatment programme, effective against F. japonica, is in place (Royal Institution of Chartered Surveyors (RICS), 2012). Such control programmes can be expensive; between £2,000 and £5,000 in total for a typical three-bedroom semi-detached house (at December 2011; Royal Institution of Chartered Surveyors (RICS), 2012). Additionally, the stigma associated with the species can result in diminution of property value (Santo, 2017) even following control action. The cumulative impact of the above is that home owners can lose all, or a significant portion, of their property’s value. This automatic restriction of mortgage options where F. japonica is present on or near a property has led to significant hardship and associated, often reported, emotional stress (Dunn, 2015; The Telegraph, 2015). The claimed ability of F. japonica to cause significant structural damage is widely acknowledged within the professional weed control sector in the UK as not being representative of the vast majority of casual field observations and that, due to current public perception, impacts on the market value of a property are out of proportion to the cost of remediation (Santo, 2017).

In order to understand if the lender response to F. japonica presence, described above, is proportionate, the impacts typically associated with F. japonica must be compared to those of other plants. The potential for plants, in general, to cause issues in the built environment is well understood. Accordingly, in the UK, developers follow guidance (NHBC, 2017) when building near trees. The automatic restriction, however, of mortgage options due to the mere presence of a plant species is a new phenomenon. Although this is currently a UK phenomenon, recent reports have emerged of F. japonica presence impacting property sales in the Republic of Ireland (C. O’Flynn, 2018, personal correspondence), suggesting that this issue has the potential to spread, and sensationalist articles have begun to appear in North American tabloids (The Calgary Eyeopener, 2015).

Plants are known to cause damage to built structures primarily by three mechanisms: (i) indirect damage, via subsidence or heave, caused by plant-mediated modifications to soil water content (Biddle, 2001; O’Callaghan & Kelly, 2005), (ii) direct damage due to physical impact, typically associated with falling trees (O’Callaghan & Kelly, 2005) and (iii) direct damage caused by physical pressure exerted through growth (Biddle, 1998, 2001).

There are many causes of subsidence, with plants only contributing to a proportion of the total and only then on shrinkable clay soils. Plant-mediated subsidence in such soils occurs when plants remove water from the soil through a process called transpiration and, as a result of this removal of water, the soil shrinks. This is particularly common during the summer months and/or periods of drought. The soil swells again once water is returned via rainfall. If foundations are not sufficiently deep or strong to withstand such stress, this process can lead to structural damage over time, typically characterised by vertical cracks up through the brickwork. Swelling of soil can also occur when mature trees, that were helping regulate soil moisture content, are removed (NHBC, 2017).

While the mechanisms behind impact-based direct damage are relatively straight forward, a range of factors—biological, chemical and physical—become relevant with respect to direct damage caused by physical pressure. Plants acquire the energy they need to grow through photosynthesis, which converts light energy, carbon dioxide and water into chemical energy that can later be released to fuel the plant’s activities. Driven by the energy produced by photosynthesis, plant roots and rhizomes grow through the soil seeking water and nutrients. Ultimately, using the products of both photosynthesis and the materials collected by roots/rhizomes, plants grow (increase in biomass) and reproduce. These growing underground plant structures follow the path of least resistance through the soil along water and/or chemical gradients, typically from areas of low water or nutrient concentration to areas of higher water or nutrient concentration (Rellán-Álvarez, Lobet & Dinneny, 2016). When solid structures (natural or anthropogenic) are encountered by extending plant tissue, highly sensitive receptors on the outer surface on the plant detect the change in pressure, resulting in the release of plant growth regulators and chemical signals that stimulate differential growth rates within plant tissues, ultimately causing the plant to grow away from the solid structure and find the path of least resistance (Takeda et al., 2008) where possible. However, where a plant becomes trapped between two structures and growth away from or around the structure is no longer possible, the risk of damage increases. The greatest risk of direct damage occurs close to the main trunk, stem or crown; this is due to incremental growth of such structures over time and secondary thickening of the roots/rhizomes, which are thickest in close proximity to such structures.

The impacts of F. japonica on residential property sale and value are ultimately predicated on the species’ ability to cause significant structural damage, but this proposition has never been scientifically tested. This paper, therefore, proposes a methodology for conducting such assessments and implements the proposed methodology using a case study of 68 residential properties in the north of England, with the aim of determining the capacity of F. japonica to cause structural damage relative to other common plants in the UK. The paper also includes an assessment of published records of F. japonica’s ability to cause structural damage; an assessment of how plants cause structural damage in the context of F. japonica’s biology; and an assessment of the findings of two surveys conducted on members of the Royal Institution of Chartered Surveyors (RICS) and the Property Care Association’s (PCA) Invasive Weed Control Group (IWCG). Additionally, given the importance of proximity, the seven-m rule is tested, based on an assessment of a survey carried out on members of the PCA’s IWCG, with the aim of determining typical rhizome extension distance relative to above ground F. japonica plants.

Materials and Methods

Study species: Fallopia japonica

Fallopia japonica is a tall, vigorous, clump-forming, herbaceous perennial, which grows up to two to three m in height (Fig. 2A) and often forms dense thickets. The stems are robust, bamboo-like, slightly fleshy and hollow, with a diameter of up to four cm. Tall-brown to bronze canes remain over winter and persist for approximately 3 years. Leaves are 10–15 cm long, lush, light green and shield-shaped with a flattened base (Fig. 2B). Growth over successive years builds up a sturdy dense crown at the base of canes (Fig. 2C). New growth primarily emerges from crowns at the start of the growth season, but also directly from rhizomes. Rhizomes are initially white, extremely fleshy and fragile while extending (Fig. 2D), but mature into yellow/orange sturdier woody structures (Fig. 2D). The majority of rhizome is found in the upper 50 cm of soil, but it can penetrate down to three m and, depending on soil type and site features, spread up to 10 m from parent plants is possible under very rare circumstances (Booy, Wade & Roy, 2015). Only female (male sterile) plants are known to be present in the UK, which form drooping grape-like clusters of flowers with distinct stigmas. Seeds are shiny, triangular, dark brown, three to four-mm long, two-mm wide and sterile in the UK. See Booy, Wade & Roy (2015) for additional information on the biology of the species. F. japonica can regenerate from rhizome fragments weighing as little as 0.7 g (Brock & Wade, 1992), providing a node is present, and from stem sections, where suitable conditions are present (very moist, well-lit soils with high nutrient availability). The species is dispersed effectively in transported soil and by water (Environment Agency, 2013; Booy, Wade & Roy, 2015). F. japonica is tolerant of a wide range of habitat and soil types, but is most frequently found in disturbed urban habitats, particularly brownfield sites, railway verges and the banks of waterways, where it thrives in damp soils.

Figure 2 Photographs illustrating F. japonica appearance and structure.

(A) F. Japonica growing within the case study area. (B) Specimen of F. japonica leaves, stem and inflorescence. (C) F. Japonica crown, associated with the plant from panel A. (D) Specimen of F. japonica mature rhizome with immature rhizomes emerging. Photos by M. Fennell.

Fallopia japonica is closely related to two other members of the Fallopia genus, F. sachalinensis and Fallopia x bohemica, which have similar invasive ranges and have similar impacts. Of note, in some parts of its invasive range, Fallopia x bohemica spreads via the production of large numbers of wind-dispersed viable seeds that germinate at rates approaching 100% in some populations (Gillies, Clements & Grenz, 2016). However, spread by this means does not currently occur in the UK.

Literature assessment

In order to contextualise impacts associated with F. japonica within the larger subject of the capacity of plants that cause structural damage, this study assessed various guidance documents and papers published on the topic of plants causing damage and the relationship between various plant traits and capacity to cause damage. The primary points of interest from these documents are highlighted in ‘Plants and structural damage’. Additionally, a focused literature search on Web of Science was conducted on 27th June 2017 to identify academic papers that provide reference to or evidence of F. japonica-mediated damage to structures. The search terms used for the Web of Science search were ‘F. japonica’ and ‘Polygonum cuspidatum’, an old name for the same species, and within the returned publications ‘damage’. The abstracts were reviewed to determine what type of damage was referred to within the paper.

Fallopia japonica impact survey

A survey of F. japonica management contractors (PCA) and property surveyors (RICS) was conducted to collect evidence either for or against the assertion that F. japonica is a major cause of structural damage to properties. Survey forms were sent out to contractors and surveyors to determine, based on their last field observation of F. japonica, the presence, if any, of damage linked to the presence of the plant across a range of built structure types (see Table 1 for included questions; see Supplemental Information S1 for individual responses). In total, 51 PCA members and 71 RICS surveyors provided records relating to 122 properties (Table 1). Each respondent was also asked how far the closest evident aboveground F. japonica plant was from the residential building on the site that they had visited. This was cross-referenced against reports of damage (Table 2). Yes/No responses are presented as raw numbers and converted to percentage values and differences between PCA and RICS respondents were considered. Statistical analyses were undertaken in PAST version 3.15 (Hammer, Harper & Ryan, 2001).

Table 1 Results from yes/no questions to contractors and surveyors.

Question	Contractor responses (n = 51)	Surveyor responses (n = 71)	
Yes	No	Yes	No	
Q1: Was there evidence of defects or structural damage to the residential building caused by the Japanese knotweed?	2% (1)	98% (50)	6% (4)	94% (67)	
Q2: Was there evidence of defects or structural damage to retaining garden walls, sheds, garages, greenhouses or lightly built garden structures caused by the Japanese knotweed?	35% (18)	65% (33)	23% (16)	77% (55)	
Q3: Was there evidence of defects or structural damage to drains, sewers and other subterranean services caused by the Japanese knotweed?	16% (8)	64% (43)	3% (2)	97% (66)	
Q4: Was there evidence of loss of amenity to the garden or grounds resulting from the presence of Japanese knotweed?	51% (26)	49% (21)	18% (13)	82% (55)	
Notes:

Results are presented as percentages for easier comparison between contractor and surveyor respondents and rounded to the nearest whole number. The actual number of responses are included in brackets. n = sample size. Three surveyors did not answer the third and fourth questions making n = 68 for those responses (see Supplemental Information S1 for more details).

Table 2 Fallopia japonica proximity to residential properties as reported by survey respondents and number of reports of damage (see Supplemental Information S1 for more details).

Distance from residential property in m	Number reported by contractors (n = 46). Reports of damage in brackets	Number reported by surveyors (n = 65). Reports of damage in brackets	
0–1.0	10 (1)	9 (3)	
1.1–2	8 (0)	3 (0)	
2.1–3	4 (0)	7 (0)	
3.1–4	2 (0)	6 (1)	
4.1–5	3 (0)	5 (0)	
5.1–6	3 (0)	1 (0)	
6.1–7	3 (0)	4 (0)	
7.1–8	2 (0)	3 (0)	
8.1–9	2 (0)	1 (0)	
9.1–10	2 (0)	8 (0)	
10.1–11	No record	1 (0)	
11.1–20	4 (0)	9 (0)	
20.1–30	2 (0)	4 (0)	
30.1–40	No record	No record	
40.1–50	No record	3 (0)	
50.1 or greater	1 (0)	1 (0)	

Fallopia japonica rhizome extent survey

The survey of PCA contractors also asked respondents to provide details, based on the last five F. japonica excavation-based remediation works that they had conducted, on the above ground area of F. japonica and to provide the horizontal (i.e. distance from visible above ground plants) and vertical (i.e. distance from soil surface) extent of rhizomes encountered. In total, 26 contractors provided records of 81 excavations with sufficient detail (e.g. clear rhizome extent linked to an identified individual stand) to be included in the assessment. Eight records were removed due to reporting multiple stands, partial excavation or disturbed sites where it was not possible to accurately determine the rhizome extent from an individual stand (see Supplemental Information S1). Subsequently, stands were sub-classified into either ‘small’ or ‘large’ categories. The small category included any plants that covered a soil area of four m2 or less, aimed at encompassing the typical size of stands found in small residential gardens. Stands covering an area greater than this were placed into the large category. This allowed for an examination of the relationship between above-ground area and rhizome extension, as well as an analysis of typical rhizome extension. Data were tested for normality (Anderson–Darling test) and differences between stand categories (large or small) were tested using the Mann–Whitney U test for non-normally distributed data. Data analyses were conducted using PAST version 3.15 (Hammer, Harper & Ryan, 2001).

Case study

A survey was conducted on 68 residential properties located on three streets in northern England. The houses on all three streets were built prior to 1900 (Consumer Data Research Centre (CDRC), 2018). All properties have been abandoned for at least 10 years and were in a state of disrepair, with most having cracked patios and crumbling brickwork (particularly on boundary walls). F. japonica was previously known to be present on properties located on all three streets. An assessment was carried out in September 2017 to determine any constraints that the species might pose to restoration and re-development (see Supplementary Information S2 for details). These sites represented a close to ‘worst case’ scenario in terms of susceptibility to damage from unchecked plant growth. With this in mind, a survey was conducted to determine presence and associated damage for F. japonica, trees, woody shrubs and woody climbers. All damage was compared against a baseline of existing damage that was present due to neglect, weathering and wear and tear over the lifetime of the properties, regardless of plant presence. Where plants were associated with damage to a structure, the damage was quantified based on the scale presented in Table 3 (see also Supplemental Information S2). Figure 3 presents examples of the rating scale that was applied.

Table 3 Scale used to quantify damage where plants were present.

Rating	Rating description	
0	Not associated with damage (e.g. just growing in soil or present beneath the soil)	
1	Correlation with existing damage (e.g. emerging from a crack in paving or a gap in brickwork, but with no detectable variation away from baseline damage)	
2	Minor exacerbation of existing damage (e.g. a detectable increase in crack width away from baseline damage)	
3	Moderate exacerbation of existing damage (e.g. a detectable addition to damage away from baseline damage, i.e. new cracks forming around an initial crack)	
4	Major exacerbation (damage beyond cracking, e.g. a damaged wall becoming undermined)	
5	Causing minor damage (e.g. creating a crack)	
6	Causing medium damage (e.g. creating a crack which has spread to form additional cracks)	
7	Causing major damage (damage beyond cracking, e.g. a previous undamaged wall becoming undermined, or concrete hard standing being significantly lifted and cracked, or a roof being smashed in due to collapse)	

Figure 3 Photographs illustrating examples of the rating scheme that was applied.

(A) Example of non-plant-based wear and tear to hard standing. (B) Rating ‘0’—B. davidii growing in a raised landscaping area, having no discernible impact on undamaged adjacent built structures. (C) Rating ‘1’—F. japonica emerging from existing cracks in paving at the base of a wall, causing no discernible impact away from baseline damage. (D) Rating ‘2’—F. japonica emerging from existing gaps in worn paving, while the gap has not been widened some mortar has been pushed aside. (E) Rating ‘3’—B. davidii growing out of a crack in worn concrete hardstanding, with additional cracks forming in the area. F. japonica visible in the background emerging from similar cracks in the hardstanding, also exacerbating existing damage but to a lesser extent. (F) Rating ‘3’—B. davidii growing out of cracks in worn brickwork, with additional cracks forming in the area. (G) Rating ‘4’—B. davidii growing out of cracks in worn brickwork. It has found its way between two structures and is facilitating the dilapidation of the wall and pushing out brickwork. (H) Rating ‘6’—B. davidii growing behind a small retaining wall and pushing some brickwork over. (I) The remains of a tree stump, which have destabilised the base of what remains of a dilapidated wall. Photos by M. Fennell.

By chance, a large number of Buddleja davidii (buddleia) plants were present at the case study sites. As such, this species was included in the assessment separately from other woody plants. B. davidii is a non-native woody shrub that is known to be invasive in the UK and elsewhere (Centre for Agriculture and Bioscience International (CABI), 2018b). Damage associated with the following species or plant groups are discussed in this case study: F. japonica, B. davidii, ‘trees’ (other woody, independently standing mature plants) and ‘woody climbers’ (woody plants that are not independently standing, e.g. attached to walls). In addition to presence, for F. japonica, mature (with crowns) and immature (without crowns) plants were assessed. Similarly, for B. davidii, mature (woody) and immature (not woody) plants were considered.

Results

Literature assessment

Plants and structural damage

The literature assessment revealed that indirect damage, typically characterised by subsidence caused by modifications to soil moisture content, was by far the most relevant mechanism identified by which plants caused major damage to built structures (Biddle, 2001; O’Callaghan & Kelly, 2005) and high water-use tree species were the most likely plant type to cause this type of damage (NHBC, 2017).

Such impacts are only a potential problem on shrinkable clay soils (Biddle, 2001; O’Callaghan & Kelly, 2005). Clay soils are found in less than 50% of the UK and not all clay soils will be equally shrinkable. The degree to which a clay soil is shrinkable depends on its mineral composition. All clay minerals are built from combinations of two types of molecular sheet, (i) a sheet with repeating units of silicon surrounded by four oxygen atoms in a tetrahedron and (ii) a sheet with an aluminium or magnesium atom surrounded by six oxygen or six hydroxyl molecules in an octahedron. How these sheets are arranged determines how ridged the clay soil is. For example, soils composed of alternating sheets, one tetrahedron followed by one octahedron, and so on, and held together by a pair of hydrogen ions are quite ridged. However, when an aluminium octahedral sheet is between two silicon tetrahedral sheets and held together by weak oxygen bonds a clay called montmorillonite is formed, which is a relatively weak clay susceptible to shrinkage (Chapman, 2012). Surveys by the Botanical Society of Britain and Ireland (Fig. 1A) show that F. japonica has been found in most areas of Britain but only a small fraction of this area is identified by the British Geological Society as having moderate to high risk of swell-shrinkage (Fig. 1B), with most shrinkable clays being found in the south east of England. Additionally, it is likely that the area at actual risk of plant-mediated shrinkage is lower again because not all of this area necessarily has the correct mineral combination required to be at high risk for facilitation of subsidence.

The second most relevant mechanism by which plants cause damage, was identified as direct damage due to physical impact, typically characterised by trees falling and striking buildings and power lines (O’Callaghan & Kelly, 2005) and is only relevant to large plants such as trees.

Finally, plants can also cause direct damage to buildings and structures by pressure exerted through growth; however, this is comparatively rare in terms of meaningful damage; it is also well understood (Biddle, 1998, 2001). While growth at the base of plants, or of roots near the surface, exerts relatively small forces, paving slabs or low boundary walls can be lifted or pushed aside. Heavy loaded or stronger structures are more likely to withstand these forces without damage, as plants preferentially distort around such obstruction before damage occurs (British Standard, 2012). Certain combinations of variables can increase the potential for damage, for example water leaking from damaged drains, sewers or water mains can encourage localised root growth, as plants typically grow towards areas of higher water availability, which can lead to roots/rhizomes entering a drain or sewer through the defect and proliferating, causing blockage and an enlarging of the initial defect. The risks associated with direct pressure based damage are (i) primarily associated with trees, (ii) vary for different types of structures and (iii) diminish rapidly with distance. Minimum recommended planting distances for young trees or new planting, to avoid direct damage to a structure from future tree growth, are described in British Standard (2012) and range from (i) no minimum distance required for planting trees near buildings, heavily loaded structures, services greater than one m deep, and masonry boundary walls, where the tree will have a stem diameter below 0.3 m (at 1.5 m above ground level) at maturity to (ii) three-m distance required for planting trees near paths and drives with flexible surfaces, paving slabs, and services less than one m deep, where the tree will have a stem diameter above 0.6 m (at 1.5 m above ground level) at maturity (British Standard, 2012).

These three mechanisms described above are evaluated against the biology and growth characteristics of F. japonica in ‘Indirect damage: in the context of F. japonica’ and ‘Direct damage: in the context of F. japonica’.

Based on the literature assessment, there is essentially no evidence to support the claim that F. japonica causes damage in excess of the norm for many plants. While evidence was found to support the claim that trees can cause major damage, no such evidence could be found for F. japonica. Of particular interest were records of insurance claims related to trees being involved in subsidence issues: 12,800 such records, between 2002 and 2005, were identified by Mercer, Reeves & O’Callaghan (2011), 1,030 of which met their criteria for records having sufficient detail to assess and as being important from a subsidence risk perspective. The top five genera implicated in subsidence-related insurance claims were Oak (Quercus), Ash (Fraxinus), Cyprus (Cupressus), Maple (Acer), and Willow (Salix). At maturity, these trees frequently reach 24, 23, 20, 18 and 24 m, respectively. No evidence of any insurance claims was identified for F. japonica with respect to structural damage. While many recent papers include in their description of F. japonica that the species can cause notable damage to built structures (Mclean, 2010; Djeddour & Shaw, 2010), this claim is never supported by evidence.

Based on the search terms ‘F. japonica’ and ‘P. cuspidatum’, the Web of Science search returned 778 journal papers published between 1937 and 2016. When the term ‘damage’ is included the number of papers dropped to 46. Five were removed for being irrelevant. Of the remaining 41 papers, 15 focused on biocontrol, 20 on general biology/genetics, two on ecological damage and two on other interactions. None of the abstracts suggested that the papers would focus on structural damage but some did refer to it as a ‘known problem’. This highlights the limited academic engagement with the problem—it appears to be accepted without supporting evidence that F. japonica causes clear and problematic structural damage.

Survey results

Survey results (reported damage)

In total, 51 contractors and 71 surveyors responded to the survey. Details of the responses are provided in Tables 1 and 2. The results of the two property damage surveys (PCA and RICS) showed clearly that reports for defects or structural damage to residential properties, where F. japonica is present, were extremely rare (between 2% and 6%). As the survey data are interpreted as a worst case situation, it is likely that more detailed surveys would reduce this number, if specifically designed to discriminate between causation, exacerbation and correlation. This statement is relevant to all types of damage reported. Reports of damage to lighter structures such as sheds or paths were more apparent, with 35% (PCA) and 23% (RICS) of respondents noticing such damage. Reports of damage to drains or subterranean services were low, 16% (PCA) and 3% (RICS). The only question to obtain a ‘yes’ above 50% was for Question 4 from the PCA contractor surveys where 51% noticed evidence for loss of amenity. However, only 18% of surveyors considered that the F. japonica observed was likely to impact garden amenity (Table 1). There was also a clear difference between the responses of surveyors and contractors for Question 3 (Table 1), with contractors reporting more damage than surveyors. It should be noted that PCA contractor members are more likely to be called out where problematic stands of F. japonica are present, which could account for the differences observed between groups. It could also be explained by differences between the two groups with respect to training, perception or bias. Investigating this was beyond the scope of the current study.

Each respondent was also asked how far the closest evident aboveground F. japonica plant was from the residential building on the site that they had visited (Table 2). This was cross-referenced (Table 2) against reports of damage, as per Question 1 (Table 1). One contractor (PCA) reported damage caused by F. japonica (Table 1); in this case the closest reported plant to the property was one m (Table 2). Four surveyors (RICS) reported damage caused by F. japonica (Table 1). Two stated that the nearest plants were zero m from the property, one stated one m from the property and one stated four m from the property (Table 2). It is worth noting that the report at four m was for a property built prior to 1900. No other responses suggested that F. japonica had caused damage to the residential property. Among contractors reporting no damage to the residential property, 25 reported F. japonica growing within four m of the residential property and a further nine reported F. japonica growing within seven m of the residential property. Among surveyors, 21 reported F. japonica within four m of the residential property and a further ten reported F. japonica within seven m of the residential property and none of these reports were linked to damage to the property. See Table 2 for more detail.

Survey results (reported rhizome extension)

There was a statistically significant difference (Mann–Whitney U; p < 0.05) in the horizontal extent of F. japonica rhizomes between small and large stands, with larger stands found to have further reaching rhizomes (Fig. 4). None of the small stands included in the assessment had rhizomes extending further than four m, and the majority (75%) had rhizomes extending two m or less. The average rhizome extension reported for small stands was 1.4 m. Only one plant in the large category had rhizome extension greater than five m (identified as a statistical outlier); all other records were below four m and the majority (75%) had rhizome extensions of 2.5 m or less.

Figure 4 Comparison of horizontal rhizome extent between small (four m2 or less) and large (greater than four m2) stands of F. japonica.

The box represents the lower 25 percentile, the median value and the upper 25% percentile and the whiskers represent the range of the data. The circle represents an outlier value (greater than two standard deviations away from the median value). Mann–Whitney U: U = 412; p < 0.05 (p = 0.01802). N = 21 (small) and 60 (large).

There was also a statistically significant difference (Mann–Whitney U; p < 0.001) between the large and small stands for vertical rhizome extent, with larger stands found to have deeper reaching rhizomes (Fig. 5). No records with vertical rhizome extent in excess of 3.5 m were recorded. The small stands had rhizomes with a mean 1.02 m depth and a maximum of two m, whereas the maximum vertical extent recorded for the large stands was 3.2 m and the mean was 1.64 m.

Figure 5 Comparison of vertical rhizome extent between small (four m2 or less) and large (greater than four m2) stands of F. japonica.

The box represents the lower 25 percentile, the median value and the upper 25% percentile and the whiskers represent the range of the data. The circle represents an outlier value (greater than two standard deviations away from the median value). Mann–Whitney U: U = 260; p < 0.0001 (p = 6.105e−5). N = 21 (small) and 60 (large).

Case study

In all but the most severe examples, the level of damage caused by plants did not exceed damage that was observed elsewhere within the study area in locations where plants were not growing. It would appear, in the context of dilapidation, that plants are generally not the cause but rather an accelerator to natural weathering and dilapidation.

Fallopia japonica was identified within the boundary of six properties (five mature stands and one immature stand) and the plant was identified within seven m of the main building of a further 12 properties, leading to a total of 18 properties where F. japonica was within the area identified by the ‘seven-m rule’ as being at risk. B. davidii was identified on 62 properties (31 mature and 31 immature). Trees were observed on six properties and woody climbers were observed on four.

In general, F. japonica was linked to less damage than the other species/species groups assessed (Table 4). Where F. japonica was linked to damage, mature plants were more likely to exacerbate the damage than to have been the original cause. There were no reported incidences of immature F. japonica causing or exacerbating damage.

Table 4 Summary data of damage linked to each of the different plant classes included in the survey.

	Plant damage to house	Plant damage to walls	Plant damage to paving	
Plants linked to damage, % of occurrences	Plants linked to damage, % of total properties	Average damage score	Plants linked to damage, % of occurrences	Plants linked to damage, % of total properties	Average damage score	Plants linked to damage, % of occurrences	Plants linked to damage, % of total properties	Average damage score	
F. japonica	0%
0/18	0%
0/68	0	11%
2/18	3%
2/68	0.029	33%
6/18	9%
6/68	0.176	
B. davidii	68%
42/62	62%
42/68	0.75	79%
49/62	72%
49.68	1.529	73%
45/62	66%
45/68	0.824	
Trees	33%
2/6	3%
2/68	0.132	67%
4/6	6%
4/68	0.235	50%
3/6	4%
3/68	0.176	
Woody climbers	75%
3/4	4%
3/68	0.103	75%
3/4	4%
3/68	0.044	0%
0/4	0%
0/68	0	
Notes:

Average damage score = the average damage value assigned to each species for each particular type of damage. For F. japonica % of properties with the species present includes those with a Knotweed plant within seven m of the main residential building (see Supplemental Information S2).

Fallopia japonica was not linked to any damage to the main buildings. The three other groups were linked to damage, at varying degrees, typically in the form of simple co-occurrence (e.g. as in appearing together without a clear causal link) or interference with brickwork through exacerbation of existing weakness. Mature woody B. davidii was more likely to exacerbate damage than immature B. davidii, with immature B. davidii rarely exceeding co-occurrence or minor exacerbation. There was only one example of a plant being linked to causing direct damage to a building, rather that exacerbating it. This was a tree falling against a house.

With respect to damage to walls, F. japonica was correlated with two occurrences of damage; in both cases it was emerging from a crack and causing no detectable variation away from baseline damage elsewhere in the wall. The three other plant groups were linked to more damage than F. japonica, to varying degrees, typically in the form of simple co-occurrence or interference with brickwork through exacerbation of existing weakness. In all groups, the average damage score was higher than that of F. japonica (Table 4). Mature woody B. davidii was more likely to exacerbate damage than immature B. davidii, with immature B. davidii rarely exceeding co-occurrence or minor exacerbation. There were only two examples of a plant being linked to causing damage to walls, rather than exacerbating it, a tree pushing over a boundary wall and B. davidii pushing over a small retaining wall.

With respect to damage to paving, F. japonica was correlated with six occurrences of damage. In three cases it was emerging from a crack and causing no detectable variation away from baseline damage elsewhere in the paving, and in three other cases it was exacerbating existing damage (one minor, two moderate examples). B. davidii was linked to more damage to paving than F. japonica, typically in the form of simple co-occurrence or interference with paving through exacerbation of existing weakness. The average damage score was considerably higher for B. davidii than F. japonica. Mature woody B. davidii was more likely to exacerbate damage than immature B. davidii, with immature B. davidii rarely exceeding correlation or minor exacerbation. There was only one example of a plant being linked to causing damage to paving, rather that exacerbating it, which was a tree where the roots had lifted a large area of concrete paving with significant associated cracking.

Discussion

Indirect damage: in the context of F. japonica

Plants are considered to cause structural damage to buildings primarily through indirect damage, for example through subsidence caused by modification to soil water content. High water-use tall trees are the main plant type implicated. Subsidence, with respect to plants, is only an issue on shrinkable clay soils, which are reasonably restricted in extent (Fig. 1). Importantly, to properly assess risk, individual site investigation is required to determine the exact type of clay present in a clay–soil area. The rate that water is removed from soil by plants varies depending on the characteristics of the plant and also by the total biomass of the plant. There is a strong linear relationship between water use and plant biomass (i.e. larger plants remove more water from the soil), as noted by Nielsen et al. (2015). Plants with higher water use and larger biomass are therefore the most likely to cause subsidence through the action of their roots removing water from soil. Some unpublished work suggests that F. japonica may be a high water use plant (Vanderklein et al., 2013); however, even if this is the case, it is not a high biomass plant by comparison to mature woody trees such as oak. The plants that are most likely to influence subsidence in the UK are listed in the NHBC (2017) guidance for building near trees. These species range in height between 10 and 28 m. In comparison, F. japonica typically only grows to between two and three m. The potential for plants to influence subsidence is calculated based on a zone of influence of between 0.5, 0.75 and 1.25 times the height of the plant (NHBC, 2017), depending on the water demand at maturity of the species in question (low, moderate or high, respectively). For F. japonica, this would suggest a maximum zone of influence of 3.75 m (the typical maximum height of the plant is three m, hence 3 × 1.25). However, when compared to mature trees, given the comparatively diminutive size of F. japonica, both in terms of above ground and below ground biomass, it is more likely to be at the lower end of the scale. As such, a calculation of 0.5 × 3 = 1.5 or 0.75 × 3 = 2.25 m is more likely to reflect the potential zone of influence of F. japonica at maturity. Furthermore, the mean rhizome length of small F. japonica stands, such as those more likely to be found in residential properties, is 1.4 m (‘Direct damage: in the context of F. japonica’ and Fig. 4), which falls comfortably within the lower zone. Such areas of influence are unlikely to be able to create a large enough area of soil shrinkage to impact all but the flimsiest of structure and, even then, only on properties shown to have shrinkable clay soil. As such, the risk associated with F. japonica causing subsidence based damage falls well below many other species commonly found in properties in the UK.

Direct damage: in the context of F. japonica

In some situations, trees and vegetation can adversely affect structures by direct action, for example structural failure of trees (collapse and impact), impact of branches with superstructures, displacement/lift/distortion and disruption of underground services and pipelines (British Standard, 2012).

The leading causes of damage due to direct physical contact by plants, that is collapsing vegetation striking buildings and power lines and branch impact, are not relevant in any meaningful way to F. japonica as the species is not tall enough and does not possess heavy enough aboveground structures. This is due to the fact that F. japonica aboveground material dies back at the end of each growth season; as such, the plant cannot accumulate sufficient above ground size and weight from successive years of growth.

Plants can also cause damage by exerting accumulating physical pressure on structures as they grow over time; however, as stated above, this is comparatively rare in terms of meaningful damage. Damage of this type is typically characterised by superficial or cosmetic damage to paving. However, more significant damage can occur where plants become trapped between two structures, for example two walls in close proximity to each other, and are allowed to exert pressure for an extended period of time without intervention (i.e. woody plants are allowed to mature in areas where management would be advisable) or where roots find their way into drains and pipes, as described above. The mechanisms by which plants grow and cause such damage are well understood (Biddle, 1998, 2001), as are the planting distances required to limit or avoid such damage (British Standard, 2012). While F. japonica can cause such damage due to direct action over time, it does not exceed that caused by woody species. The case study described in this paper demonstrates that F. japonica is less capable of causing this type of damage than trees and woody shrubs. Where F. japonica is implicated in such damage, this is likely to typically be a result of the plant exploiting a weakness or defect that was already present, rather than the plant initiating the damage, or it is simply a case of F. japonica emerging from an existing crack without influence. Regardless, even if it is assumed that F. japonica can equal trees in causing such damage (which is not the case), based on well understood principles (British Standard, 2012), a safe distance for mature F. japonica (crowns between 30 and 60 cm) would be 0.5 m for buildings and heavily loaded structures, and 1.5 m for paths and drives with flexible surfaces or paving slabs.

Additionally, the frequently stated ability of F. japonica to ‘grow through concrete’ is simply not supported by any evidence, as it is not possible due to the laws and principles of physics and biology. The extending tip of the F. japonica rhizome is remarkably soft and fleshy (Fig. 1) and it would be impossible for it to grow through intact concrete; however, these same characteristics make the extending rhizome adept at finding cracks and F. japonica has been shown to have significant ability to alter the direction of rhizome growth (Smith et al., 2007), highlighting the plant’s biological preference to go around obstructions, rather than through them. Where F. japonica is implicated in such damage, existing cracks or weaknesses are always present.

Typical rhizome extension

When the above is considered, the typical maximum rhizome extension of F. japonica is not all that relevant with respect to structural damage. Regardless, the results of the survey detailed above demonstrate that even large stands of F. japonica do not usually produce rhizomes that extend further than four m, showing that the ‘seven-m rule’ is not a statistically robust tool for estimating likely rhizome extension from above ground plants. The mean rhizome extent for small stands was 1.4 m and for large stands (above four m2) was 2.02 m. Similarly, the mean vertical extent recorded averaged between 1.02 m for the small stands and 1.64 for the large stands, with a maximum of 3.2 m.

Conclusion

The biology of F. japonica makes it less capable of causing significant structural damage than many woody plant species. This conclusion has been reached for all three of the main mechanisms by which plants are known to cause structural damage: subsidence (indirect); collapse and impact (direct); and accumulating pressure due to growth (direct). There is essentially no support for F. japonica as a major cause of damage to property in the literature, and this study found that F. japonica is less likely to cause damage than other common species. Based on the results obtained though surveys completed by PCA members, it is clear that the ‘seven-m rule’ is not a statistically robust tool for estimating likely rhizome extension. F. japonica rhizome rarely extends more than four m from above ground plants and is typically found within two m for small stands and 2.5 m for large stands. When this is considered in conjunction with the water-use requirements of an herbaceous perennial, and the limited presence of shrinkable clay soils in the UK, the likelihood of F. japonica being a major cause of structural damage decreases even further. While F. japonica is clearly a problematic invasive non-native species with respect to environmental impacts and land management, this study provides evidence that F. japonica should not be considered any more of a risk, with respect to capacity to cause structural damage in urban environments, than a range of other species of plant, and less so than many. In this context, although the impacts of F. japonica on biodiversity and other ecosystem services remain a cause for concern, there is no evidence to support automatic mortgage restriction based on the species’ presence within seven m of a building.

Supplemental Information

Supplemental Information 1 Survey results, collected from PCA and RICS contractors and surveyor, respectively.

Tab. 1: PCA member rhizome extent survey responses. Tab. 2: PCA member structural damage survey responses. Tab. 3: RICS member structural damage survey responses.

Click here for additional data file.

Supplemental Information 2 Case study site assessment results.

Tab. 1: Damage descriptors and Key for Tab. 2 nomenclature. Tab. 2: Case study damage assessment results.

Click here for additional data file.

We thank Prof Pippa Chapman, University of Leeds, for helpful discussion relating to soil properties; Chloe Spurgeon, AECOM/University of East Anglia, for supporting the literature assessment; Dr Damian Smith, AECOM, for supporting the assessment of the case study properties in the north of England; Andy Wakefield, AECOM, for support with respect to arboriculture; the Property Care Association for supporting the collection of contractor member Japanese knotweed impacts and rhizome extent data and all PCA members that provided such data; the Royal Institution of Chartered Surveyors for supporting the collection of surveyor Japanese knotweed impacts data and all RICS surveyors that provided such data; the Botanical Society of Britain and Ireland for permission to use their F. japonica map data in Fig. 1; and the British Geological Society for permission to use their shrinkable clay soil map in Fig. 1.

Additional Information and Declarations

Competing Interests

Author Contributions

Field Study Permissions

Data Availability

The authors declare that they have no competing interests. Mark Fennell (Principal Ecologist) and Max Wade (Technical Director Ecology) are employed by AECOM, UK.

Mark Fennell analysed the data, prepared figures and tables, authored and reviewed drafts of the paper, approved the final draft, designed and co-ordinated the study.

Max Wade authored or reviewed drafts of the paper, approved the final draft.

Karen L. Bacon analysed the data, prepared figures and tables, authored and reviewed drafts of the paper, approved the final draft.

The following information was supplied relating to field study approvals (i.e. approving body and any reference numbers):

The site assessment, which was carried out by AECOM ecologists, was approved via an acceptance of a scope and quote letter and an agreement of Terms and Conditions. Given the socioeconomic impacts of Japanese knotweed presence in the UK, the location and client will be kept confidential.

The following information was supplied regarding data availability:

The raw data are provided in the Supplemental Files.

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
