# Peer review of "Japanese knotweed (Fallopia japonica): an analysis of capacity to cause structural damage (compared to other plants) and typical rhizome extension"

_PeerJ, doi:10.7717/peerj.5246_

## Round 0.1 · original submission · Minor Revisions

Both reviewers found your manuscript interesting and are in favor of publication. However, there some details that you have take care before final acceptance. Please make these final adjustments and re-submit your paper. You are almost there! Good luck with your revision!

·

Basic reporting

- The text is clear, unambiguous, professional English language used
- Introduction and background is OK, literature well referenced and relevant. The spreading modes description is missing (vegetative regeneration and possible generative reproduction)
- Structure is OK
- Figures and tables are clear, high quality and relevant

Experimental design

- Original primary research on actual topic
- Questions are well defined, clear and relevant
- Methods of study are well described, but the statistical analysis used is missing.

Validity of the findings

- Validity of results is high as well as the novelty of findings
- Conclusions are well stated, linked to original research questions and limited to study results

Additional comments

I appreciate the topic of the study and serious research in this field. I suggest to the authors add the topic of presence of other taxa of the genus Fallopia, mainly the hybrid F. ×bohemica, which can cause similar damages as F. japonica. I also miss the mention of regeneration ability form rhizomes and possible generative spread (within the genus) – my suggestion is to add least short paragraph on this topic.
The description of statistical analysis used is missing for both, the surveys and the rhizome case study. This is the only weakness of the manuscript and it should be added before Acceptance.

Reviewer 2 ·

Basic reporting

The manuscript, "Japanese knotweed (Fallopia japonica): An analysis of capacity to cause structural damage (compared to other plants) and typical rhizome extension" is interesting and novel. However, I feel that before publication there are major edits to the language and the organization of the manuscript that are required. Please proof read to remove awkward long sentences and redundancy.
Specific comments:
Line 53: flood not floor
Line 63: There is no mention of when knotweed first came to the UK. Consider moving lines 156-164 to the introduction.
Line 84: What is a knock-on impact?
Lines 120-130: This description of basic physiology of plants is unnecessary.
Lines 135-137: Awkward
Line 180-182: Missing word?
Line 468-471: Redundant. This has been described more than once already.

Experimental design

METHODS
Literature Assessment:
Expand the search to include damage by other plants. At a minimum by B. davidii, as it is highlighted in another section and in a table.
Impact Survey:
Did surveys include questions about other plant species?
Rhizome Extent Survey:
Include information on how the data was analyzed.
RESULTS
Literature Assessment:
You describe the results of a literature review that is not described by your methods. It reads like a literature review that is not appropriate for a results section.
Line 260: "This variable was investigated further" This process should be described in the methods, not the results.
Lines 263-276: There is an unnecessary level of detail about soils.
Table 4. Replace "Ivy" with "Woody Climbers," or at least be consistent with the text in the body of the manuscript.

Validity of the findings

I agree with and appreciate the novel findings presented by this manuscript. However, the description of the methods and the clarity of the discussion should be improved. The manuscript is at times redundant.
Line 359: Why would a house built prior to 1900 indicate that the damage is correlation/exacerbation and not causation? This conclusion is not clear.
Beginning at line 401: You go into great depth with the B. davidii comparison. Please provide more description of this plant in an earlier section.

---

## Round 0.2 · accepted · Accept

I believe that the authors have addressed all suggestions made by the reviewers in a satisfactory manner. Therefore, I am accepting this manuscript in its present form. Congratulations!

#